# Improvement of Electrical Performance in P-Channel LTPS Thin-Film Transistor with a-Si:H Surface Passivation

**DOI:** 10.3390/ma12010161

**Published:** 2019-01-07

**Authors:** Kyungsoo Jang, Youngkuk Kim, Pham Duy Phong, Younjung Lee, Joonghyun Park, Junsin Yi

**Affiliations:** College of Information and Communication Engineering, Sungkyunkwan University, 2066 Seobu-ro, Jangan-gu, Suwon-si, Gyeonggi-do 16419, Korea; jks30716@skku.edu (K.J.); bri3tain@skku.edu (Y.K.); pdphong@skku.edu (P.D.P.); younjlee@paran.com (Y.L.)

**Keywords:** poly-Si TFT, LTPS TFT, FT-IR, Raman, surface passivation, leakage current

## Abstract

We report the effects of surface passivation by depositing a hydrogenated amorphous silicon (a-Si:H) layer on the electrical characteristics of low temperature polycrystalline silicon thin film transistors (LTPS TFTs). The intrinsic a-Si:H layer was optimized by hydrogen dilution and its structural and electrical characteristics were investigated. The a-Si:H layer in the transition region between a-Si:H and µc-Si:H resulted in superior device characteristics. Using a-Si:H passivation layer, the field-effect mobility of the LTPS TFT was increased by 78.4% compared with conventional LTPS TFT. Moreover, the leakage current measured at V_GS_ of 5 V was suppressed because the defect sites at the poly-Si grain boundaries were well passivated. Our passivation layer, which allows thorough control of the crystallinity and passivation-quality, should be considered as a candidate for high performance LTPS TFTs.

## 1. Introduction

Currently, the display industry has been widely investigating to meet new market demands for low-cost, large-size, high-resolution, high-frame-rate, and 3D displays. Thin film transistors (TFTs) as switching or driving devices are one of the most important components. Low temperature polycrystalline silicon (LTPS) TFTs have been widely used because their field effect mobility is higher than that of hydrogenated amorphous silicon (a-Si:H) TFTs, so that high resolution displays with ultra-high densities (3840 × 2160, UHD) can be fabricated. Because poly-Si TFTs are very robust under electrical and optical stress, they can be used in various applications [1]. However, there are crucial disadvantages of LTPS TFTs, such as high off-state leakage currents originating from defects in the grain boundaries of poly-Si [2]. To reduce leakage currents in poly-Si TFTs, some additional structures have been proposed including lightly doped drain (LDD) and offset gate structures [3]. However, these structures require additional processing and costs, so they have not been fully applied in the display industry. To reduce the off-state leakage current, Kim et al. proposed a simple method based on the insertion of a thin (10 nm) a-Si:H layer [4]. They explained the reason for the reduced leakage current using an energy band diagram and the current paths in the LTPS TFT with an a-Si:H passivation layer and a larger band gap. Inserting a larger band gap material between the active layer and gate insulator can result in a reduced rate of thermionic emission. Thus, the minimum leakage current was successfully reduced and the on/off ratio was improved. However, the field-effect mobility (12 cm^2^/V·s) of the LTPS TFT with an a-Si:H passivation layer was remarkably lower than that of a TFT without an a-Si:H passivation layer (58.3 cm^2^/V·s). Although they solved the leakage current problem in LTPS TFTs through the simple insertion of a passivation layer with a larger band gap, the field-effect mobility was severely decreased, so that the high performance of poly-Si TFTs disappeared.

In this work, we investigated the reduction of leakage current without sacrificing the field-effect mobility. We deposited an a-Si:H passivation layer on the poly-Si layer. An a-Si:H passivation layer has been used to achieve a high conversion efficiency in hetero junction solar cells [5,6]. However, the use of an a-Si:H passivation layer in poly-Si TFTs has so far not been reported. The optimized a-Si:H layer passivates dangling bonds at the interface and reduces carrier recombination, thus improving the device characteristics. We investigated the deposition of an a-Si:H passivation layer and optimized the dilution gas ratio. Different a-Si:H layers were deposited using different gas ratios of SiH_4_:H_2_, and the characteristics of the a-Si:H layers were evaluated by Raman spectroscopy, Fourier transform infrared spectroscopy (FT-IR), and quasi-steady-state photoconductance (QSSPC) measurements to examine their crystallinity (X_C_), defects, bonding, and passivation-quality. Finally, LTPS TFTs with different passivation layers were fabricated and their electrical performance was evaluated. The leakage current was successfully reduced without sacrificing the field-effect mobility by using an optimized a-Si:H layer.

## 2. Device Fabrication

A 50-nm-thick a-Si:H film was deposited on the 300-nm-thick buffer oxide layer by plasma enhanced chemical vapor deposition (PECVD). The a-Si:H layer was dehydrogenated at 400 °C in a N_2_ atmosphere for 10 min and then crystallized using an XeCl (308 nm) excimer laser. After the formation of poly-Si, the intrinsic a-Si:H passivation layer was deposited on the poly-Si. To investigate the effect of surface passivation on the electrical performance of LTPS TFTs, the 10-nm-thick passivation layers were deposited by PECVD using different silane (SiH_4_) and hydrogen (H_2_) gas flow rates. Various dilution gases, such as helium (He) and hydrogen, have already been studied for the deposition of a-Si:H [7]. The layer of a-Si:H deposited using He as a dilution gas was found to contain disordered and loose networks. However, the a-Si:H deposited with H_2_ dilution gas was found to form denser structure, so we used H_2_ as the dilution gas in this work. The SiH_4_ flow rate was varied from 20 to 6 sccm and the H_2_ flow rate was varied from 20 to 36 sccm. The gas ratio (GR) of H_2_/(SiH_4_ + H_2_) was changed from 0.50 to 0.86, while the plasma power density was fixed at 1.27 W/cm^2^. The deposition temperature for the passivation layers was 180 °C, and the RF power was 100 W, while the deposition pressure was 50 mTorr. The highest and lowest deposition rates of a-Si:H passivation layers were 2.59 nm/min at GR of 0.75 and 0.55 nm/min at GR of 0.5, respectively. The 220-nm-thick SiO_2_ gate insulator was deposited by PECVD at 180 °C on the passivation layer. An aluminium layer was deposited on the gate insulator to serve as the gate metal, and then the substrate was patterned. Source and drain doping were performed using a boron ion shower for the p-type TFT. Then, the source and drain regions were activated at 400 °C in a N_2_ atmosphere for 4 h. The channel width and length of the fabricated TFTs were 180 and 50 µm, respectively. The electrical characteristics of the LTPS TFTs with different passivation layers were examined using a semiconductor parameter analyzer at room temperature under dark conditions. To further understand the passivation layer’s structural and electrical characteristics, Raman spectroscopy, Fourier transform infrared spectroscopy (FT-IR), and quasi-steady-state photoconductivity (QSSPC) measurements were conducted.

## 3. Results and Discussion

Raman spectroscopy is a well-known technique for distinguishing the structural characteristics of thin films. In particular, it can indicate the ordering of an amorphous silicon film. The defects of a-Si:H are mainly detected at low frequency (300~400 cm^−1^). We investigated the use of Raman spectroscopy for evaluating the passivation layer. Figure 1a shows the Raman spectroscopy of passivation layers deposited using different GR values. The 100-nm-thick passivation layer was deposited on a glass substrate. The measured Raman spectroscopy were decomposed into several gaussian component peaks located at around 150, 300, 410, and 480 cm^−1^, corresponding to the transverse acoustic (TA), longitudinal acoustic (LA), longitudinal optical (LO), and transverse optical (TO) modes [8,9,10]. The Raman characteristics of the passivation layers were strongly influenced by the GR. When the GR was increased from 0.5 to 0.75, the ratios of TA/TO, LA/TO, and LO/TO were reduced. When the GR was over 0.75, the ratios of TA/TO, LA/TO, and LO/TO rapidly increased. The increase of TA/TO, LA/TO, and LO/TO has been related to the enhanced generation of defects in passivation layers [8,9,10]. Figure 1b shows the ratios of TA/TO, LA/TO, and LO/TO at different GR values. Furthermore, the crystallinity (X_C_) of the passivation layer was calculated using the Raman peak ratio of the broad band at 480 cm^−1^ (a-Si:H) and the strong band at 520 cm^−1^ (c-Si). A Raman spectrum characteristic of µc-Si:H was observed at GR values above 0.75 and the X_C_ sharply increased.

To further understand the silicon hydrogen bonds, we performed FT-IR measurements, which can easily detect the incorporation of Si and H. To analyze the FT-IR, a 100-nm-thick passivation layer was deposited on c-Si. Figure 2a,b shows the microstructure parameter (R*) of the passivation layers deposited by different GR values. The FT-IR absorption of a-Si:H and µc-Si:H provides specific information about the type of bonds present between Si and H [11,12]. In this study, two kinds of bonds with a stretching vibrational mode were observed to investigate the structural properties of the passivation layers, such as Si-H bonds with a peak at around 2000 cm^−1^ and Si = H_2_ bonds with a peak at around 2090 cm^−1^. As the GR was increased to 0.75, I_2000_ also increased. However, I_2000_ decreased for GR values over 0.8. To calculate the properties of the passivation layer, the quantity, R*, was introduced. R* was defined as I_2000_/(I_2000_ + I_2090_). The Si = H_2_ bonding factor representing the quality of the layer was regarded as a measure of the defects or voids. As Si = H_2_ bonding increased, many defects remained in the thin film. The R* value gradually increased from 0.27 at a GR of 0.5 to 0.45 at a GR of 0.75 and sharply decreased to 0.15 at a GR of 0.86. Based on the correlation with Raman results, the highest R* value and the transition region from a-Si:H to µc-Si:H were well matched.

To investigate the characteristics of different passivation layers, QSSPC measurements were conducted for a-Si:H on a single crystalline silicon wafer (c-Si) (a-Si:H/c-Si), µc-Si:H on c-Si (µc-Si:H/c-Si), and a pure c-Si sample. Figure 2c shows the carrier lifetime characteristics of the passivation layers formed using different passivation layers. This measurement is widely used in the contactless characterization of solar cells. Many thin films, such as a-Si:H and Al_2_O_3_, have been employed to passivate the interface or bulk defects [13,14]. The system is based on a radio frequency bridge with an inductive coil that generates electromagnetic fields in the wafer. The interfaces of a silicon substrate represent a severe discontinuity in its crystalline structure. The poly-Si, especially at the interface, has many dangling bonds, so that a large density of defect levels could be found within the bandgap near the interface. By using an appropriate surface passivation layer, such as an optimized a-Si:H film, these dangling bonds in poly-Si grain boundaries could be passivated, thus reducing the interface trap density [15]. A flash lamp was irradiated on a semiconductor surface. The excessive electron and hole pairs were generated, and then generated carriers were decayed. The minority carrier lifetime (τ) is the amount of time during which photo-generated carriers remain free, highly mobile in the material, and at the surface, before recombining in some defect sites [16]. Because the τ of a thin film could be dominantly influenced by surface conditioning, and it can be useful for evaluating the passivation quality. In this study, c-Si substrate without a passivation layer was used as a reference with τ value of 22.23 µs. The τ of a-Si:H/c-Si with GR of 0.75 was the highest at 51.77 µs and the τ of µc-Si:H/c-Si with GR over 0.75 was reduced below the reference τ. By using the optimized a-Si:H (GR = 0.75) passivation layer, the lowest interface trapped charge density was expected.

Figure 3 shows the electrical properties obtained from transfer curves of the p-channel LTPS TFTs with and without passivation layers on the poly-Si layer. The width and length of the p-channel LTPS TFTs were 180 µm and 50 µm, respectively (Width/Length = 3.6). The transfer curves were measured at a drain bias of −0.1 V. The extracted electrical properties of the LTPS TFTs with and without passivation layers are displayed in Table 1. The field effect mobility (*µ*) was calculated by the maximum transconductance method at V_DS_ = −0.1 V and the threshold voltage was observed at the (Width/Length) × V_GS_ at drain current density of 10 nA. The leakage current (*I_D.Leak_*) was measured at a V_GS_ of 5 V. The LTPS TFT without a passivation layer had the following electrical properties: *µ* of 49.58 cm^2^/V·s, subthreshold swing (*S.S.*) of 0.91V/dec, and the *I_D.Lea__k_* of 7.62 × 10^−11^ A/cm^2^. However, when the optimized passivation layer (GR =0.75) was employed on poly-Si layer, the LTPS TFT exhibited high *µ* of 88.53 cm^2^/V·s, S.S of 0.58V/dec and *I_D.Leak_* of 2.46 × 10^−12^ A/cm^2^. Moreover, the threshold voltage was considerably increased. These improved TFT characteristics were attributed to the fact that the optimized a-Si:H layer can easily passivate the poly-Si interface with high trap densities. Especially, it was known that the improved threshold voltage (*V_TH_*) and *S.S.* are related to deep defect states. The characteristics of poly-Si TFTs fabricated at a low temperature were dominated by interface and grain boundary defect states. It was clear that the amount of trap states between the poly-Si layer and the gate oxide layer was reduced due to the optimized a-Si:H layer, as proven by FT-IR and QSSPC measurements. The leakage current was also reduced by the passivation layer. Significant band-bending occurs between the channel and drain region because of the reversely biased p-n junction, where the leakage current can flow via the defect sites at the poly-Si grain boundary [17]. The optimized passivation layer was effective to reduce the number of such defect sites. To express this numerically, the interface defect sites between SiO_2_ and poly-Si were estimated by the Levinson and Proano method [18,19]. The number of defect sites can be expressed as:(1)$$N_{T}=\left[\left(\frac{S.S.}{ln10}\right)\left(\frac{q}{kT}\right)-1\right)]\left(\frac{C_{ox}}{q}\right)$$
where *S.S.* is the subthreshold swing, *q* is the unit charge, *T* is the absolute temperature, *k* is the boltzmann constant and $C_{ox}$ is the capacitance of the gate oxide.

The technology computer aided design (TCAD) simulation was conducted to understand the defect states distribution in the LTPS TFT. The characteristics of LTPS TFT can be modeled by the distribution of the density of states (DOS) in the band gap. In the case of p-type LTPS TFT, the on current and field effect mobility was affected by the density of the donor like tail state defects (NTD) near the valance band, while the threshold swing and threshold voltage was affected by the donor like deep state defects (NGD). The transfer characteristics of LTPS TFT was fitted in TCAD simulation. The LTPS TFT without a passivation layer had NTD of 9.9 × 10^12^/eVcm^3^ and NGD of 7.7 × 10^12^/eVcm^3^. The LTPS TFT with the optimized a-Si:H passivation layer had NTD of 9.9 × 10^11^/eVcm^3^ and NGD of 2.3 × 10^12^/eVcm^3^. Additionally, the LTPS TFT with the µc-Si:H passivation layer had NTD of 9.9 × 10^13^/eVcm^3^ and NGD of 2.9 × 10^13^/eVcm^3^.

The number of interface defect states was successfully reduced by using a passivation layer. However, the on current (V_GS_ < −10 V) characteristics were quite different. In the case of LTPS TFTs with µc-Si:H passivation layers, the electrical properties were degraded with higher *S.S* and lower field-effect mobility. The most likely reason for this degradation is the creation of new dangling bonds on the poly-Si layer by highly diluted hydrogen. Our passivation process can supply additional hydrogen for the passivation, but it could also create new dangling bonds [20]. Therefore, the dilution gas ratio for the passivation layer was carefully controlled to avoid creating new dangling bonds.

In the LTPS TFT, various defect states in the grain boundaries and intra-grain influence the electrical characteristics as well as the carrier transport from the source to the drain. The poly-Si is often terminated at the interface imperfectly. The trap states at the grain boundaries are associated with the lattice discontinuities by differently oriented grains. The a-Si:H passivation layer supplies hydrogen atoms combined with silicon, and it can passivate dangling bonds in poly-Si. When the passivation layer was deposited at low values of GR, the hydrogen is predominantly incorporated in the mono-hydrogen (Si–H) bonding. However, as GR is increased, the bonding shifts from mono-hydrogen (Si–H) to di-hydrogen (Si–H_2_) [12]. We think that a proper mixture of Si-H and Si-H_2_ in the optimized a-Si:H layer might be a good passivation quality.

Figure 4 shows a schematic diagram of the energy bands of p-channel LTPS TFTs with and without passivation layers under negative and positive gate bias based on the energy band difference, defect density, and current paths. When a negative gate voltage was applied to the LTPS TFTs, the energy band diagram is represented in Figure 4, (a) for ‘only poly-Si’, (b) ‘a-Si:H/poly-Si’, and (c) µc-Si:H/poly-Si. Compared to the diagram in (a), the a-Si:H/poly-Si layers resulted in a bandgap difference between a-Si:H (E_g_ = 1.88 eV) and poly-Si (E_g_ = 1.12 eV), and the channel carrier mobility was enhanced due to the improved interface quality. In the case of µc-Si:H (E_g_ = 1.2 eV)/poly-Si, the effect of this bandgap difference was smaller, resulting in a degraded interface quality. When the gate voltage was positive, the energy band diagram can be represented as in Figure 4, (d) for ‘only poly-Si’, (e) ‘a-Si:H/poly-Si’, and (f) µc-Si:H/poly-Si. In the case of (d), when the gate voltage was positive, the carriers flowed from the drain to the source via poly-Si. In the case of the poly-Si with passivation layer, when the gate voltage was positive, the carriers flowed from the drain to the source via the passivation layer and/or passivation layer/poly-Si/buffer. The field emission, denoted by path C, for the SiO_2_/poly-Si TFT was suppressed by the presence of a wide band gap passivation. Therefore, the dominant current path may be A or B as depicted in Figure 4d–f [21]. Path A represents the thermionic field emission in the passivation region and path B represents the thermionic field emission in the poly-Si due to the weakened electric field in the poly-Si by the presence of the passivation layer. 

## 4. Conclusions

We fabricated p-channel LTPS TFTs with a-Si:H passivation layer to reduce the leakage current without sacrificing field effect mobility. Poly-Si TFTs have excellent and reliable electrical characteristics, but they still suffer from leakage currents under off state bias. To suppress the leakage current, many techniques have been tried, but the field-effect mobility of the poly-Si TFT was severely decreased. In this work, a thin a-Si:H passivation layer with the wide band gap and high passivation-quality was deposited on poly-Si layer. The a-Si passivation layer was optimized by controlling the dilution gas ratio and its structural and electrical characteristics were reported. The on state and off state characteristics of the poly-Si TFTs with optimized a-Si:H passivation layer were considerably improved because the defect sites at the poly-Si grain boundaries were well passivated. Based on the findings of this study, the use of the passivation layer with high passivation-quality should be considered as potential candidates for high performance LTPS TFTs.

## Figures and Tables

**Figure 1 materials-12-00161-f001:**
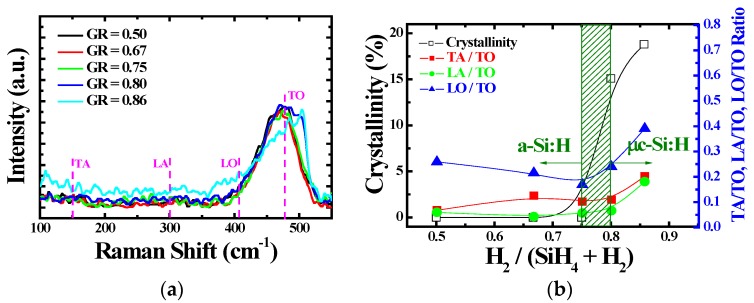
(**a**) The Raman spectroscopy of passivation layers as a function of gas ratio (GR) (**b**) The dependence of the Raman characteristic parameters and crystallinity on GR.

**Figure 2 materials-12-00161-f002:**
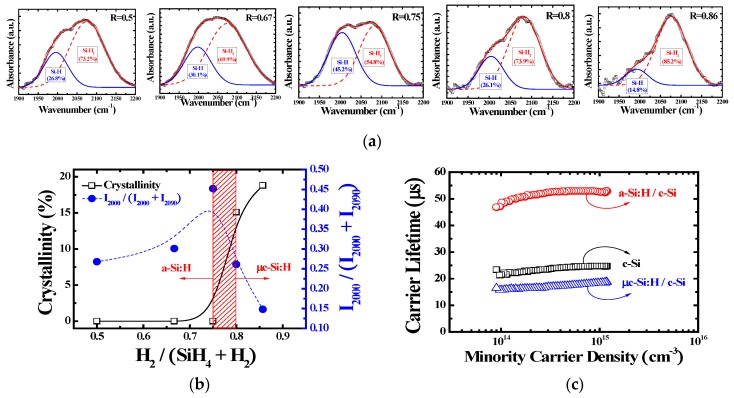
(**a**) The Fourier transform infrared (FT-IR) spectroscopy characteristics as a function of GR (**b**) The calculated crystallinity by Raman spectroscopy and microstructure by FT-IR (**c**) Carrier lifetime characteristics of the passivation layers formed using different passivation layers by QSSPC.

**Figure 3 materials-12-00161-f003:**
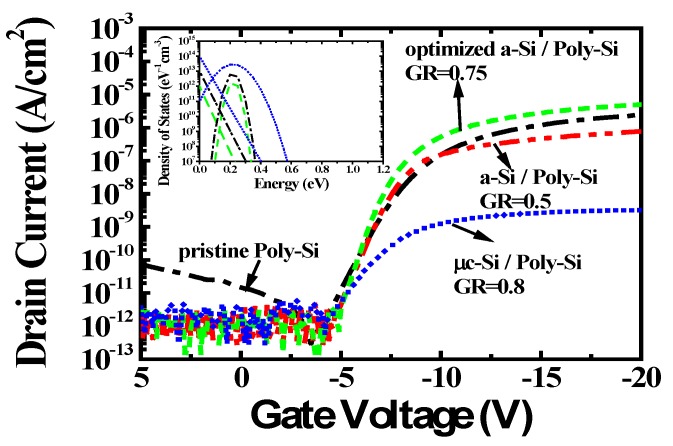
Transfer characteristics of low temperature poly-Si thin film transistors (LTPS TFTs) with and without passivation layers. The inset figure shows the defect states in the LTPS TFTs simulated by technology computer-aided design (TCAD).

**Figure 4 materials-12-00161-f004:**
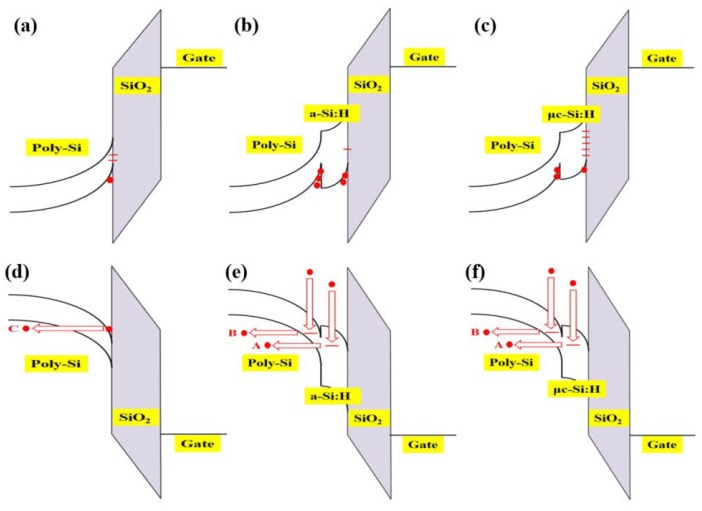
The schematic energy band diagram of p-channel LTPS TFTs with and without passivation layers under negative and positive gate bias based on the energy band difference, defect density, and current paths. (**a**) Only poly-Si under negative gate bias, (**b**) a-Si:H/poly-Si under negative gate bias, (**c**) µc-Si:H/poly-Si under negative gate bias, (**d**) only poly-Si under positive gate bias, (**e**) a-Si:H/poly-Si under positive gate bias, and (**f**) µc-Si:H/poly-Si under positive gate bias.

**Table 1 materials-12-00161-t001:** Comparison of electrical characteristics of p-channel LTPS TFTs with and without passivation layers on glass substrates.

Parameter	Without Passivation Layer	With Passivation Layer		
	Poly-Si	a-Si:H (GR = 0.5)	Opt. a-Si:H (GR = 0.75)	µc-Si:H (GR = 0.8)
*µ* (cm^2^/Vs)	49.58	18.2	88.53	1.3
*S.S.* (V/dec)	0.91	0.72	0.58	1.19
*N_T_* (cm^−2^)	7.38 × 10^12^	5.78 × 10^12^	4.62 × 10^12^	9.71 × 10^12^
*V_TH_* (V)	−6.75	−6	−5.9	−6.4
*I_d.leak_* (A/cm^2^)	7.62 × 10^−11^	2.3 × 10^−12^	2.46 × 10^−12^	3.68 × 10^−12^

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
