# Peer review of "Improvement of Electrical Performance in P-Channel LTPS Thin-Film Transistor with a-Si:H Surface Passivation"

_materials, 2019, doi:10.3390/ma12010161_

Reviewer 1 Report

Submitted manuscript (Materials-391360) on "Improvement of Electrical Performance in P-Channel LTPS Thin-Film Transistor with a-Si:H Surface Passivation" is well written. It also documents sufficient data/analysis to demonstrate mobility improvement and passivation evidences to justify for implementation of a-Si layer in LTPS TFTs. This is an excellent work of technical merit and scientific value to researchers in field of display Device design, specifically passivation materials/layers where understanding of electrical & structural characteristics response to field effects and defect site interaction is  critical to design high performance LTPS TFTs. Here are few comments to author to further improve this work:

- English and grammar need significant improvement throughout this manuscript.
- Abstract & introduction is concise and well written. However, additional references may be included as plenty of research discusses advantages of a-Si:H as passivation layers.
- Specify temperature, RF freq and chamber pressure for PECVD a-Si:H passivation layer deposition.
- Include device image and microstructural evidence of film morphology as gas dilution ratio changes. Elaborate further on deep defect states (type, capture cross-section, quantitative variation for different GRs).
- Fig3 - Change mC-Si to u-Si.

Please add a section to discuss mechanism for preferred a-Si:H layer properties that would drive for more appropriate passivation response.

Overall this works shows excellent progress and promise for improved understanding of passivation layer evaluation.  

Author Response

Dear reviewer,

We appreciate you for your comments helping us to modify the manuscript (materials-391360) better for the readers of Materials.

Herewith, we summarize the revisions we made for the reviewer’s comments. Please read the below through; we believe that we did our best as possible as we can to follow every comments of the review.

 We colored statements responding to reviewer’s comments in blue only to help the reviewer’s convenience. And, The red color statements are revised statements in manuscript. We hope these statements will turn in black at the final publication.

Please find attached documents.

Best regards,

Junsin Yi

College of Information and Communication Engineering,

Sungkyunkwan University,

Suwon, Gyeonggi-do, Republic of Korea

Tel: +82-31-290-7139

Email: junsin@skku.edu

Reviewer 2 Report

The authors report on the passivation properties of a hydrogenated amorphous silicon (a-Si:H) layer deposited on low temperature polycrystalline silicon thin film transistors (LTPS TFTs). The intrinsic a-Si:H is a well-known passivation layer for heterojunction solar cells structure, and such passivation layer was not reported yet for a poly-Si TFTs structure, which underlines the originality of this work. The characterizations performed are relevant with the effective lifetime by the quasi steady-state photoconductance (QSSPC). Moreover the authors carefully study the different phases between amorphous silicon to microcrystalline silicon, by FTIR and Raman spectroscopy, pointing out the importance of the deposition conditions and more precisely the gas ratio (GR) of H2 to decrease the interface state density. Then, electrical characterization such as the transfer characteristics of LTPS TFTs are presented to underline the advantage of such passivation layer and justify the optimal deposition condition with an enhanced carrier mobility.

However some major points must be taken into consideration:

-       Concerning the QSSPC measurement, the explanation of the minority carrier lifetime and its measurements (lines 123-124, and lines 129-132) are not convincing. The lines 132-133 are not correct, the lifetime τ is not “the amount of time that generated carriers remain at defect sites of the Si surface”. On the contrary, a more accurate description would be: the lifetime is the amount of time during which photo-generated carriers remain free, highly mobile in the material and at the surface, before recombining in some defect sites. During the transition between “photo-generated free carriers” and the equilibrium “without photo-generated carriers” the resistivity of the silicon wafer is modified, which is measured by the QSSPC equipment. The authors are encouraged to carefully read about the Sinton equipment, which is the mostly used for the determination of carrier lifetime on silicon wafer, and also to add the following reference [1] for an accurate description of the QSSPC measurement.

[1]      R. A. Sinton and A. Cuevas, “Contactless determination of current–voltage characteristics and minority‐carrier lifetimes in semiconductors from quasi‐steady‐state photoconductance data,” Appl. Phys. Lett., vol. 69, no. 17, pp. 2510–2512, 1996.

-       During the device fabrication, concerning the 220-nm-thick SiO2 gate insulator which is deposited directly on the passivation layer: the author could mention the deposition process and the deposition temperature of such thick SiO2 layer to underline that the crystallinity and the microstructure factor of the passivation layer are not disturbed.

-       The authors could comment on the thickness difference between the 10-nm-thick passivation layer from the device fabrication, compared to the 100-nm-thick passivation layer used for Raman spectroscopy and FTIR analysis?

-       Line 172, what is the “on characteristics”? It is probably related to the “on state and off state characteristics” found in the conclusion. However those terms should be clarified earlier inside the result and discussion part.

Beside, several minor points have also to be considered:

-       Considering the carrier lifetime, it is strongly depending on the excess carrier density, also referred as the injection level. The lifetime values are commonly reported at 1×1015 cm-3. For accuracy, the authors could also mention at which injection level the lifetime values are reported? For example with the reference value of τ at 22.23 µs (line 134).

-       In the device fabrication part, the a-Si:H as passivation layer is probably not doped, if it is true, the authors should explicitly mentioned it as an “intrinsic a-Si:H”.

-       The term “p-channel LTPS TFTs” appears in Line 184, but could be briefly developed inside the device fabrication section, is it also a p-channel used for the Fig. 3? If so, it should be explicitly mentioned in the text.

-       The expression “interface trap state” (line 129), is commonly often reported as “interface trap density” or “interface state density”.

-       In the equation 1, even if it is obvious, the use of the Boltzmann constant k should be mentioned. Also, what is the value of Cox used here (the capacitance of the gate oxide)? Does the Cox value remain identic for all samples?

-       Line 198, the authors probably want to refer to Fig. 4 (e) and (f) only.

-       Some English formulation: sentences (lines 199-200-201) and (lines 212-213-214) should be reformulate to avoid confusion.

-       The English could be polished slightly. For example, some minor points suggested:

o   Line 19: “should be considered as a candidate”

o   Line 24: “widely investigating to meet”

o   Line 47: “We deposited an a-Si:H passivation layer”

o   Line 106: “deposited at different GR values”

o   Line 111: “ To calculate the properties”

o   Line 118: “To investigate the properties of different passivation layers”

o   Line 191: “… was smaller, resulting in a degraded interface quality”

o   Line 193,195: “ when the gate voltage is positive”

 After the consideration of the above mentioned major and minor points, I would recommend this work to be published in Materials journal, as the experiments and characterizations clearly present novel and interestingly results to the scientific community.

Author Response

Dear reviewer,

We appreciate you for your comments helping us to modify the manuscript (materials-391360) better for the readers of Materials.

Herewith, we summarize the revisions we made for the reviewer’s comments. Please read the below through; we believe that we did our best as possible as we can to follow every comments of the review.

 We colored statements responding to reviewer’s comments in blue only to help the reviewer’s convenience. And, The red color statements are revised statements in manuscript. We hope these statements will turn in black at the final publication.

Please find attached documents.

Best regards,

Junsin Yi

College of Information and Communication Engineering,

Sungkyunkwan University,

Suwon, Gyeonggi-do, Republic of Korea

Tel: +82-31-290-7139

Email: junsin@skku.edu

Response to Reviewer 2 Comments

 The authors report on the passivation properties of a hydrogenated amorphous silicon (a-Si:H) layer deposited on low temperature polycrystalline silicon thin film transistors (LTPS TFTs). The intrinsic a-Si:H is a well-known passivation layer for heterojunction solar cells structure, and such passivation layer was not reported yet for a poly-Si TFTs structure, which underlines the originality of this work. The characterizations performed are relevant with the effective lifetime by the quasi steady-state photoconductance (QSSPC). Moreover the authors carefully study the different phases between amorphous silicon to microcrystalline silicon, by FTIR and Raman spectroscopy, pointing out the importance of the deposition conditions and more precisely the gas ratio (GR) of H2 to decrease the interface state density. Then, electrical characterization such as the transfer characteristics of LTPS TFTs are presented to underline the advantage of such passivation layer and justify the optimal deposition condition with an enhanced carrier mobility.

However some major points must be taken into consideration:

-       Concerning the QSSPC measurement, the explanation of the minority carrier lifetime and its measurements (lines 123-124, and lines 129-132) are not convincing. The lines 132-133 are not correct, the lifetime τ is not “the amount of time that generated carriers remain at defect sites of the Si surface”. On the contrary, a more accurate description would be: the lifetime is the amount of time during which photo-generated carriers remain free, highly mobile in the material and at the surface, before recombining in some defect sites. During the transition between “photo-generated free carriers” and the equilibrium “without photo-generated carriers” the resistivity of the silicon wafer is modified, which is measured by the QSSPC equipment. The authors are encouraged to carefully read about the Sinton equipment, which is the mostly used for the determination of carrier lifetime on silicon wafer, and also to add the following reference [1] for an accurate description of the QSSPC measurement.

[1]      R. A. Sinton and A. Cuevas, “Contactless determination of current–voltage characteristics and minority‐carrier lifetimes in semiconductors from quasi‐steady‐state photoconductance data,” Appl. Phys. Lett., vol. 69, no. 17, pp. 2510–2512, 1996.

- Thanks for kind suggestion. Description about the QSSPC was revised.

The minority carrier lifetime (τ) is the amount of time during which photo-generated carriers remain free, highly mobile in the material and at the surface, before recombining in some defect sites [16].

-       During the device fabrication, concerning the 220-nm-thick SiO2 gate insulator which is deposited directly on the passivation layer: the author could mention the deposition process and the deposition temperature of such thick SiO2 layer to underline that the crystallinity and the microstructure factor of the passivation layer are not disturbed.

- The deposition condition was revised in the paper.

The 220-nm-thick SiO2 gate insulator was deposited by PECVD at 180 °C on the passivation layer.

-       The authors could comment on the thickness difference between the 10-nm-thick passivation layer from the device fabrication, compared to the 100-nm-thick passivation layer used for Raman spectroscopy and FTIR analysis?

- Thanks for kind comment. We think that thickness effect on Raman and FT-IR can be occurred. When the thin film thickness is increase, Raman & FT-IR peak intensity can be increased. Because the µc-Si is grown as a columnar, the thickness effect can be occurred.

But, Raman & FT-IR peaks of 10nm thick passivation layer were uncertain in our laboratory, so that we tried to evaluate thicker layer. Thus, the evaluation of buffer layer was experimented at 100nm thick layer. But, the thickness of each of buffer layer is identical to compare characteristics.

-       Line 172, what is the “on characteristics”? It is probably related to the “on state and off state characteristics” found in the conclusion. However those terms should be clarified earlier inside the result and discussion part.

- Thanks for kind suggestion. The description for on characteristics was revised in the paper.

The number of interface defect states was successfully reduced by using a passivation layer. But, the on current (VGS < -10V) characteristics were quite different.  

Beside, several minor points have also to be considered:

 -       Considering the carrier lifetime, it is strongly depending on the excess carrier density, also referred as the injection level. The lifetime values are commonly reported at 1×1015 cm-3. For accuracy, the authors could also mention at which injection level the lifetime values are reported? For example with the reference value of τ at 22.23 µs (line 134).

- Thanks for kind suggestion. In our laboratory, the injection level was about 1e16/cm3. You can find the measurement data at else paper.

“Effect of valence band offset and surface passivation quality in the silicon heterojunction solar cells”  DOI:10.1149/2.031111jes

-       In the device fabrication part, the a-Si:H as passivation layer is probably not doped, if it is true, the authors should explicitly mentioned it as an “intrinsic a-Si:H”.

- Thanks for kind suggestion. The description for a-Si:H layer was revised in the paper.

After the formation of LTPS, the intrinsic a-Si:H passivation layer was deposited on the LTPS. To investigate the effect of surface passivation on the electrical performance of LTPS TFTs, the 10-nm-thick passivation layers were deposited by PECVD using different silane (SiH4) and hydrogen (H2) gas flow rates.

-       The term “p-channel LTPS TFTs” appears in Line 184, but could be briefly developed inside the device fabrication section, is it also a p-channel used for the Fig. 3? If so, it should be explicitly mentioned in the text.

- Thanks for kind suggestion. The boron doping for p-type TFT was inserted in the device fabrication section.

Source and drain doping were performed using a boron ion shower for the p-type TFT.

-       The expression “interface trap state” (line 129), is commonly often reported as “interface trap density” or “interface state density”.

- The expression “interface trap state” was revised to “interface trap density”.

By using an appropriate surface passivation layer, such as an optimized a-Si:H film, these dangling bonds in poly-Si grain boundaries could be passivated, thus reducing the interface trap density [15].

-       In the equation 1, even if it is obvious, the use of the Boltzmann constant k should be mentioned. Also, what is the value of Cox used here (the capacitance of the gate oxide)? Does the Cox value remain identic for all samples?

- The description for equation 1 was revised.  The used Cox was 3.4×10-8 F/um2. The Cox value is remain identical because the characteristics of gate oxide was not changed.

where S.S. is subthreshold swing, q is unit charge, T is absolute temperature, k is boltzmann constant and Cox is the capacitance of the gate oxide.

-       Line 198, the authors probably want to refer to Fig. 4 (e) and (f) only.

- Thanks for kind suggestion. The proper reference for Fig.4 was added.

21. Kim, J.; Sohn, K.; Jang, J.; Temperature dependent leakage currents in polycrystalline silicon thin film transistors. IEEE transection Electron Devices 1990, 37, 1727-34. DOI: 10.1063/1.365416

-       Some English formulation: sentences (lines 199-200-201) and (lines 212-213-214) should be reformulate to avoid confusion.

- The manuscript was revised.

Path A represents the thermionic field emission in the passivation region and path B represents the thermionic field emission in the poly-Si due to the weakened electric field in the poly-Si by the presence of the passivation layer.

In this work, thin a-Si:H passivation layer with a wide band gap and high passivation-quality was deposited on poly-Si layer.

-       The English could be polished slightly. For example, some minor points suggested:

 o   Line 19: “should be considered as a candidate”

- The manuscript was revised.

Our passivation layer, which allows thorough control of the crystallinity and passivation-quality, should be considered as a candidate for high performance LTPS TFTs.

o   Line 24: “widely investigating to meet

- The manuscript was revised.

Currently, the display industry has been widely investigating to meet new market demands for low-cost, large-size, high-resolution, high-frame-rate, and 3D displays.

o   Line 47: “We deposited an a-Si:H passivation layer”

- The manuscript was revised.

We deposited an a-Si:H passivation layer on the poly-Si layer.

o   Line 106: “deposited at different GR values”

- The manuscript was revised.

Fig. 2(a) shows the microstructure parameter (R*) of the passivation layers deposited by different GR values.

o   Line 111: “ To calculate the properties”

- The manuscript was revised.

To calculate the properties of the passivation layer, the quantity R* was introduced. R* was defined as I2000 / (I2000 + I2090).

o   Line 118: “To investigate the properties of different passivation layers

- The manuscript was revised.

To investigate the characteristics of different passivation layers, QSSPC measurements were conducted for a-Si:H on a single crystalline silicon wafer (c-Si) (a-Si:H/c-Si), µc-Si:H on c-Si (µc-Si:H/c-Si), and a pure c-Si sample.

o   Line 191: “… was smaller, resulting in a degraded interface quality”

- The manuscript was revised.

In the case of µc-Si:H (Eg=1.2eV) /poly-Si, the effect of this bandgap difference was smaller, resulting in degraded interface quality.

o   Line 193,195: “ when the gate voltage is positive”

- The manuscript was revised.

When the gate voltage is positive, the energy band diagram can be represented as in Fig. 4 (d) for ‘only poly-Si’, (e) ‘a-Si:H/poly-Si’, and (f) µc-Si:H/poly-Si.

 After the consideration of the above mentioned major and minor points, I would recommend this work to be published in Materials journal, as the experiments and characterizations clearly present novel and interestingly results to the scientific community.

- We would like to appreciate your review.

Reviewer 3 Report

This manuscript describes the effect of a-Si insertion layer between LTPS channel and gate insulator. The TFT properties with the optimized a-Si:H interfacial layer were increased compared to that without the layer. The authors claim that this might be caused by decrease in interfacial trap density, thus carrier lifetime increased and mobility enhanced. I agree this clear mechanism. However, the thin film characterization by raman is very unclear. Although the data is important in this work and is necessary to discuss the effect of the quality of the interfacial layer, the difference of TA, LA, LO peaks is hard to recognize. Unfortunately, I feel that the manuscript is not suitable for publication at this moment because of the lack of data reliability and estimation.

Author Response

Dear reviewer,

 We appreciate you for your comments helping us to modify the manuscript (materials-391360) better for the readers of Materials.

Herewith, we summarize the revisions we made for the reviewer’s comments. Please read the below through; we believe that we did our best as possible as we can to follow every comments of the review.

 We colored statements responding to reviewer’s comments in blue only to help the reviewer’s convenience. And, The red color statements are revised statements in manuscript. We hope these statements will turn in black at the final publication.

Please find attached documents.

Best regards,

Junsin Yi

 College of Information and Communication Engineering,

Sungkyunkwan University,

Suwon, Gyeonggi-do, Republic of Korea

Tel: +82-31-290-7139

Email: junsin@skku.edu

Response to Reviewer 3 Comments

This manuscript describes the effect of a-Si insertion layer between LTPS channel and gate insulator. The TFT properties with the optimized a-Si:H interfacial layer were increased compared to that without the layer. The authors claim that this might be caused by decrease in interfacial trap density, thus carrier lifetime increased and mobility enhanced. I agree this clear mechanism. However, the thin film characterization by raman is very unclear. Although the data is important in this work and is necessary to discuss the effect of the quality of the interfacial layer, the difference of TA, LA, LO peaks is hard to recognize. Unfortunately, I feel that the manuscript is not suitable for publication at this moment because of the lack of data reliability and estimation.

- We would like to appreciate your review.

The Fig.1 is the Raman spectra of passivation layer as a function of GR. In our laboratory, the intensity of Raman was too noisy to recognize for reviewer. We apologize this figure. But, we convince that the trend of TA/TO, LA/TO and LO/TO at different GR. As the GR was increased from 0.5 to 0.75, the value of TA, LA, LO and TO was almost same, Moreover, the ratio of peak was very low. At the GR of 0.8, the peak of TO was slightly increased. On the other hand, the peak of TO was rapidly decreased at the GR of 0.86. Thus, we make sure the Raman spectra in the paper.
